# The Effect of Synonymous Single-Nucleotide Polymorphisms on an Atypical Cystic Fibrosis Clinical Presentation

**DOI:** 10.3390/life11010014

**Published:** 2020-12-27

**Authors:** Giovana B. Bampi, Anabela S. Ramalho, Leonardo A. Santos, Johannes Wagner, Lieven Dupont, Harry Cuppens, Kris De Boeck, Zoya Ignatova

**Affiliations:** 1Biochemistry and Molecular Biology, Department of Chemistry, University of Hamburg, 20146 Hamburg, Germany; giovanabampi@gmail.com (G.B.B.); leonardo.santos@chemie.uni-hamburg.de (L.A.S.); LitusWagner@t-online.de (J.W.); 2CF Organoids Research Lab, Woman and Child Unit, Department of Development and Regeneration, University of Leuven, 3000 Leuven, Belgium; anabela.santoramalhovenancio@kuleuven.be (A.S.R.); harry.cuppens@skynet.be (H.C.); christiane.deboeck@uzleuven.be (K.D.B.); 3Cystic Fibrosis Center, University Hospital of Leuven,3000 Leuven, Belgium; lieven.dupont@uzleuven.be; 4Department of Pediatrics, Pediatric Pulmonology, University Hospital of Leuven, 3000 Leuven, Belgium

**Keywords:** cystic fibrosis, synonymous SNP, case report, homozygous c.1584G>A, p.Gln528= patient-derived organoids

## Abstract

Synonymous single nucleotide polymorphisms (sSNPs), which change a nucleotide, but not the encoded amino acid, are perceived as neutral to protein function and thus, classified as benign. We report a patient who was diagnosed with cystic fibrosis (CF) at an advanced age and presented very mild CF symptoms. The sequencing of the whole cystic fibrosis transmembrane conductance regulator (*CFTR*) gene locus revealed that the patient lacks known CF-causing mutations. We found a homozygous sSNP (c.1584G>A) at the end of exon 11 in the *CFTR* gene. Using sensitive molecular methods, we report that the c.1584G>A sSNP causes cognate exon skipping and retention of a sequence from the downstream intron, both of which, however, occur at a relatively low frequency. In addition, we found two other sSNPs (c.2562T>G (p.Thr854=) and c.4389G>A (p.Gln1463=)), for which the patient is also homozygous. These two sSNPs stabilize the CFTR protein expression, compensating, at least in part, for the c.1584G>A-triggered inefficient splicing. Our data highlight the importance of considering sSNPs when assessing the effect(s) of complex CFTR alleles. sSNPs may epistatically modulate mRNA and protein expression levels and consequently influence disease phenotype and progression.

## 1. Introduction

Typically, genes directly linked to disease have been identified through nonsense or missense mutations (i.e., mutations that introduce a premature stop codon or exchange an amino acid, respectively), leading to nonfunctional protein [1]. In this context, since most of the amino acids are encoded by more than one codon triplet, synonymous single nucleotide polymorphisms (sSNPs) in the protein-coding part of the gene which exchange a nucleotide, but not the encoded amino acid are considered as benign with no discernable effect on the disease phenotype [2]. Emerging evidence challenges this “neutrality for function” view on sSNPs. sSNPs increase the repertoire of genetic variability, causing alterations of mRNA stability, miRNA-mediated gene regulation, mRNA translation velocity and consequently change protein expression levels and activity [3,4,5,6,7,8,9]. Moreover, sSNPs that are located at and/or in close proximity of exon boundaries may also alter splicing signals. An extensive and systematic mutation analysis of exons 10 and 13 of the *CFTR* gene presents a large set of sSNPs that may alter splicing [10].

Here, we describe an atypical CF patient who is homozygous for the c.1584G>A sSNP (rs1800095, p.Glu528=), an sSNP located at the boundary of exon 11. This sSNP was first identified in a 5-year old CF patient of Celtic origin [11] and an Arabic patient [12]. In the general population, the heterozygous c.1584G>A sSNP is relatively frequent, with an allele frequency of 1.7%, but it is much rare as homozygous sSNP (<0.3%) [13]. In 200 Belgian CF-chromosomes, the A-allele displayed 2.5% frequency (5 alleles from a total of 200) [14]. In a Spanish CF cohort, the c.1584G>A sSNP was prevalent (22%) among idiopathic chronic pancreatitis (ICP) patients and suggested as a predisposing factor for chronic pancreatitis [15]. The c.1584G>A variant has also been reported to be enriched in patients with recurrent acute pancreatitis [16]. In contrast, other studies report that this sSNP is less enriched in patients with diffuse bronchiectasis than in the control group [17]. Based on epidemiologic data, the c.1584G>A sSNP is classified as a neutral and non-CF causing variant; however, cell culture experiments show alterations of CFTR protein and mRNA levels and hence classify this sSNP as a non-neutral variant in vitro [17]. With patient-derived material (colorectal organoids) and in in vitro cell culture models, we characterized the *CFTR* mRNA expression pattern and found that along with the full-length transcript, a minor fraction of the *CFTR* transcript underwent aberrant splicing, including both exon 11 skipping and intron 11 retention. In addition, this patient is also homozygous for two other sSNPs, i.e., c.2562T>G and c.4389G>A, which enhance the CFTR protein expression, and thus likely counteracting the losses from the partially inefficient splicing.

### 1.1. Case Presentation

In this case report, we present a female patient who, at a later age (66 years), was diagnosed with very mild CF symptoms, pancreatic sufficiency, and abnormal sweat chloride levels. Two sweat tests showed sweat chloride levels of 73 and 84 mmol/L, respectively. The patient had been suffering from recurrent respiratory infections since early adulthood and was diagnosed with bronchiectasis at the age of 48 years and of seropositive rheumatoid arthritis (RA) at the age of 64 years; the bronchiectasis deemed as unlikely to be due to RA because of the large time span between two diagnoses. She was treated with a low dose of oral steroids (e.g., methotrexate and subsequently adalimumab). The patient was referred to the CF Center in Leuven because of recurrent respiratory exacerbations at the age of 66 years. No known CF-causing mutation associated with CF was found in the first round of screening. She presented symptoms of chronic mucopurulent bronchitis, diffuse cylindrical and cystic bronchiectasis (more pronounced in the right middle and upper lobe) with partial atelectasis of the right middle lobe, as well as an obstructive pulmonary function (FVC 3.06 L or 121% pred and FEV_1_ 1.51 L or 72% pred). Her sputum culture grew intermittently *Staphylococcal aureus*, non-mucoid *Pseudomonas aeruginosa,* and *Aspergillus fumigatus*; the specific IgG for *Pseudomonas* was negative, and NTM cultures were repeatedly negative. She had an IgE and an IgG sensitization to *Aspergillus fumigatus* without a formal diagnosis of ABPA (total IgE was 119 kU/L). The patient presented no gastrointestinal symptoms. Her BMI was 19.8, blood lipids, vitamin A, D, E levels and PTT were normal (blood cholesterol, 191 mg/dL, triglycerides, 99 mg/dL, vitamin A, 377 µg/L, 1.25-dihydroxyvitamin D, 46.5 µg/L, vitamin E, 15.2 mg/L, PTT, 104% or INR, 1.1), and her fecal elastase was also normal (453 µg/gr feces). IgG, IgG_1,2,3_, IgA, IgM levels were normal, nasal NO was normal.

Colorectal organoids derived from the patient displayed a wild-type-like morphology and, at baseline, were already swollen with a basal luminal area similar to that of the non-CF controls. The sequencing of the whole *CFTR* gene revealed a homozygous c.1584G>A sSNP (p.Glu528=) located at the 3’ edge of exon 11 and hence, the only indication of potential pathogenicity. Furthermore, the patient is homozygous for two other sSNPs (e.g., c.2562T>G (p.Thr854=) and c.4389G>A (p.Gln1463=)) and heterozygous for many common, moderately common, and rare variations in the non-coding regions, none of which alone is associated with CF.

### 1.2. Molecular Analysis of the Effect of the c.1584G>A sSNP

The c.1584G>A sSNP is at position 1584, the last nucleotide of exon 11 and exchanges codon GAG to GAA, both encoding Glu at position 528 in CFTR (p.Glu528=). Nucleotides close to the splicing junction modulate correct splicing, thus using the *CFTR* gene sequence, we first predicted in silico the effect of c.1584G>A sSNP using various online tools. Five different tools predict a c.1584G>A sSNP-induced decrease of the splicing strength at the boundary between exon 11 and intron 11 (Table 1), and hence, suggest that the c.1584G>A sSNP would cause some splicing defects.

Next, we used patient-derived colorectal organoids, from which we extracted total RNA and subjected it to RT–PCR and total mRNA-sequencing (RNA-Seq). The RT–PCR analysis supported the skipping of exon 11 (Figure 1A), which was only observed in the organoids of the patient harboring the c.1584G>A sSNP, but not in organoids of another patient with CF-causing mutation (F508del/3272-26A) suggesting the causality of the c.1584G>A sSNP for the exon 11 skipping (Figure 1A).

In addition, we employed RNA-Seq of the patient’s organoids. A close inspection of the coverage of the *CFTR* exon 11—intron 11 borders revealed sequencing reads mapping to the intronic region of the patients’ *CFTR* transcript (Figure 1B, right panel), suggesting intron retention. Notably, this effect is not detected in the healthy control (Figure 1B, left panel), suggesting the causality of the c.1584G>A sSNP for intron retention. It should be noted that RNA-Seq in this form of analysis cannot detect exon skipping. In sum, the *CFTR* transcript analyses with two independent methods provide evidence for aberrant splicing events leading to both exon 11 skipping and intron 11 retention.

We next employed an in vitro system to address the effect of the c.1584G>A sSNP on CFTR protein expression. This experimental setup allows determining the effect on protein level without the contribution of the aberrant mRNA splicing. We introduced the c.1584G>A sSNP into the CFTR cDNA and expressed it in CFBE41o^-^ cells. To our surprise, compared to the wild-type CFTR the c.1584G>A sSNP enhanced by appr.1.3-fold the protein expression level of the fully glycosylated functional CFTR form (band C, Figure 2A). Since the patient is homozygous for two additional sSNPs, c.2562T>G (p.Thr854=) and c.4389G>A (p.Gln1463=), we also introduced them, along with the c.1584G>A sSNP. This combination greatly augmented the expression of the mature form of CFTR (Figure 2C).

### 1.3. Transcriptome-Wide Effect of the c.1584G>A sSNP

Next, to assess the effect of c.1584G>A sSNP on a cell-wide level in more representative settings, we used RNA-Seq and compared the *CFTR* transcript expression profile in the p.Glu528 patient-derived organoids and compared them to organoids derived from a non-CF subject. *CFTR* transcripts exhibited much higher expression level in the c.1584G>A sSNP organoids (49 and 42 rpM in each independent replicate) compared to the organoid from the non-CF subject (16 rpM), which also mirrors the differences in protein expression level in CFBE41o^-^ cells (Figure 2A).

RNA-Seq also allows for comparison of the total gene expression—an approach proven recently to be powerful in determining the molecular signature of mutation-associated CF pathology [18]. Unlike lung-derived material, colorectal organoids are derived from an infection-free environment and are thus suitable to address whether mutation-triggered changes in *CFTR* expression alone elicit stress response [18]. We detected approximately 6000 transcripts in the organoid samples and compared their expression between each independent replicate of the c.1584G>A sSNP organoids with the organoid of the healthy non-CF individual (using log_2_(fold change) cut-off of +/−2 and *p* < 0.05). Small sets of transcripts were upregulated and downregulated (i.e., from 62 to 377, respectively; Figure 2B,C). A cross-comparison between both independent replicates of the c.1584G>A sSNP organoids showed that a large fraction of the observed gene alterations captured the natural heterogeneity of independent replicates (Figure 2C). In total, we found 32 and 108 transcripts that were uniquely and reproducibly upregulated and downregulated in the c.1584G>A sSNP organoids, respectively (Figure 2C). Gene ontology analysis did not reveal any functional clustering of the upregulated or downregulated transcripts; rather, punctual functionalities were influenced. In addition, among the upregulated genes, we did not detect genes from stress response categories, i.e., inflammatory, unfolded protein response stress, or tissue remodeling pathways recently reported as a signature of severe CF pathologies [18]. Taken together, these results suggest that the homozygous c.1584G>A sSNP caused only marginal alterations of the global gene expression with no indication of the activated molecular stress response, which on a cellular-wide level corroborates the very mild CF phenotype of this patient.

## 2. Discussion

Here we analyzed the effect of sSNPs in a subject with a very mild atypical CF phenotype. The c.1584G>A sSNP, for which the patient is homozygous, is not alone CF-causing. Since it exchanges the last nucleotide of exon 11, it has been suggested to cause splicing aberrancies [19]; however, the molecular mechanism was unclear. Using highly sensitive approaches, including deep sequencing of the *CFTR* transcript, we show that the c.1584G>A sSNP causes both intron 11 retention and exon 11 skipping, however, at a relatively low extent so that the larger portion of the *CFTR* transcript is likely accurately processed. The observed mild alterations in global gene expression with no detectable activation of stress or inflammatory pathways corroborate on a molecular level the very mild CF phenotype of this patient. Possibly, the aberrantly spliced *CFTR* transcripts may accumulate over time, which could be an explanation for the late disease onset and late-age of diagnosis of the patient.

Splicing mutations are usually detected relatively late because of difficulties to clearly uncover their effect and/or assign a causal link to disease. In general, various splicing mutations in the *CFTR* have been associated with mild CF phenotypes [20,21,22]. Previous studies have reported the presence of c.1584G>A in cohorts with CF or CF-like symptoms [11,12,14,15,16,17], although the prevalence of this sSNP is also similar to that in healthy individuals [19]. Hence, far, this is the first report of a patient homozygous for the c.1584G>A sSNP. The homozygosity could amplify the effect of this sSNP, because of the gene dosage effect, compared to an individual compound heterozygous for it. The patient presents a very mild and rather atypical CF phenotype. While the positive sweat test, respiratory features and chronic pulmonary infections resemble the CF symptoms, the mild effect of the c.1584G>A sSNP on *CFTR* expression along with the wild-type morphology of the patient-derived organoids is a feature of a typical non-CF mutation. The patient also bears a rare combination of two other homozygous alleles, i.e., c.2562T>G, c.4389G>A, which are protective and positively influence the CFTR protein expression level [5,23]. Likely, they also contribute to this very mild and atypical CF clinical presentation.

Our results highlight the importance of assessing the effect(s) of complex alleles, i.e., of concomitantly occurring variants, on mRNA and protein expression levels in order to explain more complex and unusual CF disease phenotype and progression. This raises the awareness that sSNPs affecting the nucleotide sequence, but not the amino acid sequence and, thus, currently classified as benign, may contribute to CF pathology. Furthermore, this study emphasizes the ability of sensitive transcriptomics approach to detect rarely occurring splicing events and to assess differences in gene expression signatures caused by a mutation. In the near future, this technology could be implemented to deeply characterize rare and complex CF alleles with yet unknown pathology, and potentially inform personalized approaches to clinical intervention.

## 3. Materials and Methods

### 3.1. Sequencing of the Complete CFTR Locus

A region of 500 kb, including the complete *CFTR* gene with introns and flanking intergenic regions (e.g., promoter), were targeted for sequencing by highly parallel sequencing. The average sequencing read depth was 393-fold, and 99.5% of the nucleotides in the covered *CFTR* regions had a minimum read depth of 10 reads. The following kits were used: *CFTR* locus enrichment kit: HaloPlex (chr7:117027043-–117527041) (coverage 96.60%; Agilent Technologies, Santa Clara, CA, USA), HiSeq rapid PE Cluster kit v2 or HiSeq PE Cluster Kit v4 cBot, and HiSeq SBS Kit V4, 200 cycli (HiSeq 2500), or MiSeq Reagent Kit v2 (MiSeq, Illumina, San Diego, CA, USA). Bioinformatic analysis was performed on CLC Genomics Workbench 9.0.1 (CLC Bio, Qiagen, Red Wood City, CA, USA) using the reference sequence and annotations: H. sapiens, hg19, GRCh37 (February 2009; 1-based). Variants with a read count percentage above 77% were concluded as homozygous, and those with a read count percentage between 23 and 77% as heterozygous. Variants at nucleotide positions with insufficient sequence reads depth, or large deletions/duplications in heterozygous that are difficult to detect in highly parallel sequencing assays may be missed in the probabilistic mutation calling approach. Variants were confirmed by Sanger sequencing.

### 3.2. Organoid Culture

Biopsies were collected from a 69-years old subject homozygous from the CFTR variant c.1584G>A. Crypts were isolated from rectal biopsies collected as described earlier [24]. The crypts were cultured with Matrigel under specific medium conditions in order to produce a rectal organoid culture from this patient [24]. The organoids were expanded and collected.

### 3.3. RT–PCR

To determine whether the c.1584G>A sSNP alter the exon11-intron 11 splicing, we used a qualitative RT–PCR based approach. RNA was extracted from the organoids (collected from 3 wells of a 24-well plate after growing for one week) using the RNAspin Mini RNA isolation kit (GE Healthcare, Wauwatosa, WI, USA). 0.5 µg total RNA was used for cDNA using random hexamers and SuperScript IV reverse transcriptase (Invitrogen) a final volume of 20 µL. The region spanning exon 10-exon12 was PCR amplified with forward B5R (5′-AACTTCTAATGGTGATGACAGCCT-3′) and reverse EX11L (5′-TATATTGTCTTTCTCTGCAAACTTGG-3′) primer and GoTaq DNA polymerase (Promega, Madison, WI, USA). Organoids from a patient with CF-causing mutation (F508del/3272-26A) were used as a CF control.

### 3.4. Organoid RNA-Sequencing

Total RNA was extracted from p.Glu528= rectal organoids (2 independent replicates, i.e., independently grown organoids from the patient) and organoids from individual with no CF-related mutations using the TRIzol-reagent (Sigma-Aldrich, Taufkierchen, Germany), and the protocol followed as described [18]. The expression levels of each transcript were presented as mapped reads normalized to the total sequencing reads (i.e., reads per million, rpM).

### 3.5. Cell Culture and Transfection

Parental CFBE41o^-^ cells (kind gifts of Karl Kunzelmann, University of Regensburg, Germany; Dieter Gruenert, University of California San Francisco, USA) were maintained in Earle’s minimal essential medium (MEM; PAN Biotech, Aidenbach, Germany), supplemented with 2 mM L-glutamine (Gibco) and 10% fetal calf serum (FCS; PAN Biotech, Aidenbach, Germany). Wild-type or c.1584G>A variant cDNAs—alone or in combination with other sSNPs—were cloned into the pcDNA3 vector (ThermoFisher, Waltham, MA, USA) and transfected with polyethyleneimine (linear, MW 40.000 Da, Polysciences, Hirschberg an der Bergstrasse, Germany).

### 3.6. Antibodies and Immunoblotting

The following CFTR antibodies were employed: mouse anti-NBD2 (596; dilution 1:2000), all of which were purchased from John R. Riordan and Tim Jensen (University of North Carolina, Chapel Hill, NC, USA) through the Cystic Fibrosis Foundation Therapeutics Antibody Distribution Program (CFF, Bethesda, MD, USA). Additional antibodies utilized in this study include mouse anti-β-actin (anti-ACTB; dilution 1:4000; Sigma-Aldrich, Taufkirchen, Germany, no. A228); rabbit anti-NPT (dilution 1:1000; Millipore, no. H06-747); goat anti-mouse-HRP (dilution 1:10,000; Bio-Rad, Feldkirchen, Germany, no. 170-5047) and goat anti-rabbit-HRP (dilution 1:3000; Bio-Rad, Feldkirchen, Germany, no. 170-5046).

Western blots with CFBE41o- were performed as annotated before [5]. For normalization, intensities of CFTR bands B and C were normalized to (i) neomycin phosphotransferase (NPT), which is expressed within the pcDNA3 backbone and therefore provides a measure of transfection efficiency, and (ii) β-actin (ACTB), which serves as sample loading controls.

### 3.7. Exon-Skipping Prediction

The following tools were used to predict in silico the strength of the splicing site scores in wild-type and in CFTR bearing the c.1584G>A sSNP: NetGene2 Server [25]; NNSPLICE [26]; Hbond [27]; MaxEntScan [28], alternative splice site predictor (ASSP) [29], and SpliceAI [30]. The splicing score reflects the strength of the splice site and is defined independently by each program. The scores are summarized in Table 1.

## Figures and Tables

**Figure 1 life-11-00014-f001:**
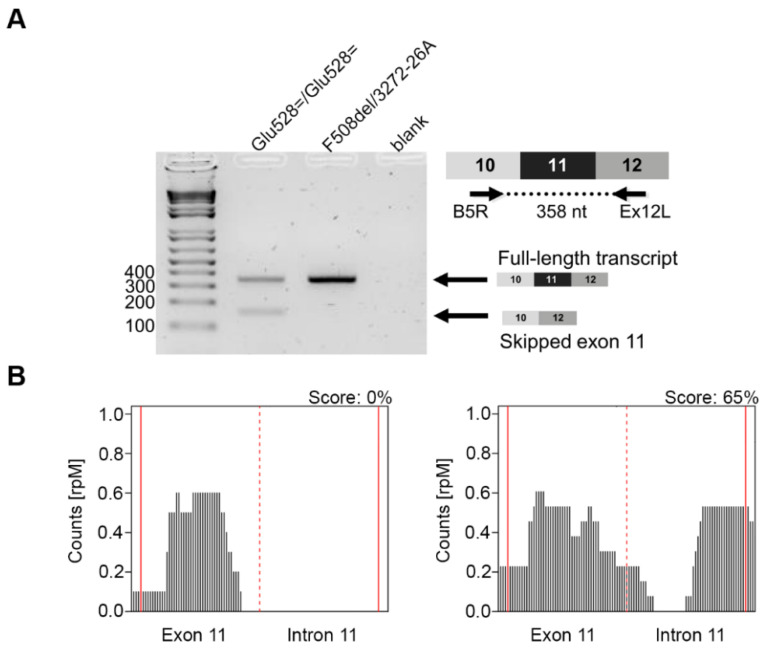
c.1584G>A sSNP induces splicing aberrancies. (**A**) Skipping of exon 11 detected in the p.Glu528= patient-derived organoids by qualitative RT–PCR. Exon 11 is 192 bp long. An amplicon lacking exon 11 is 166 bp (lower band) in the p.Glu528= homozygous organoids. The numbers on the left denote the size of the fragment in nt of the 1kb-DNA ladder. (**B**) Retention of intron 11 detected in the p.Glu528= patient-derived organoids by RNA-Seq. Sequencing coverage of the *CFTR* transcript at the junction of exon 11 and intron 11 in the organoids of a non-CF subject (left panel) and the p.Glu528= patient-derived organoids (right panel). Read counts are presented in 50-nt windows from both exon 11 and intron 11 (solid red lines) to the intron/exon junction (dashed red line). Genuine intron coverage is selected from sporadic coverage events by an arbitrary coverage score ≥of 20% (top of each plot). This score is defined as Σ_intron 50 nt window_ (reads coverage)/Σ_exon 50 nt window_ (reads coverage) × 100%.

**Figure 2 life-11-00014-f002:**
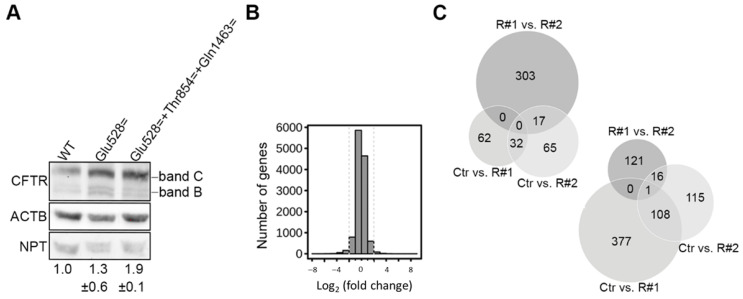
c.1584G>A sSNP augments CFTR protein expression and causes marginal alterations in the global gene expression. (**A**) Representative immunoblots of CFTR variants transiently expressed in CFBE41o^-^ cells and probed with anti-CFTR NBD2 antibody (CFF#596). Band B and band C denote immature and mature CFTR glycoforms, respectively. Neomycin phosphotransferase (NPT), encoded on the same plasmid, served as an internal transfection control, whereas β-actin (ACTB) was utilized as loading control. WT, wild-type. The numbers under the blot represent a quantification C/(C + B) normalized to NPT and ACTB. Data are means ± SEM. (**B**) Fold-change of independent RNA-Seq replicate of the p.Glu528= patient-derived organoids (n = 2) compared to organoids from a non-CF individual. (**C**) Venn diagrams comparing the upregulated (right diagram) and downregulated genes (left diagram) between each independent replicate of the p.Glu528= organoids (R#1, R#2) and non-CF organoids (Ctr). The variability between the replicates (R#1 vs. R#2) is also included; Pearson’s coefficient, R^2^ = 0.92, indicating high reproducibility between the two replicates.

**Table 1 life-11-00014-t001:** In silico prediction of the splicing strength at the boundary between exon 11 and intron 11 for wild-type cystic fibrosis transmembrane conductance regulator (CFTR) and CFTR harboring the c.1584G>A synonymous single nucleotide polymorphisms (sSNPs). For all predictive tools, higher scores reflect the higher strength of the splice site at the end of exon 11.

Variant	Algorithm				
	NatGene2 ^a^	NNSplice ^a^	Hbond ^b^	MaxEntScan	ASSP
wt	0.93	1	17.3	10.06	12.19
c.1584G>A	0.89	0.95	12.6	6.42	9.53

^a^ The score is defined between 0 and 1. Score of 1 implies the highest confidence of the true splicing site. ^b^ Higher score implies a stronger capability of forming H-bonds with U1 snRNA.

## Data Availability

Sequencing data from the healthy colon organoids were deposited within Gene Expression Omnibus (GEO) under accession number GSE143621. Sequencing from two replicates of p.Glu528= patient-derived organoid is accessible in GEO under accession number GSE154023.

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
