# Peer review of "The Effect of Synonymous Single-Nucleotide Polymorphisms on an Atypical Cystic Fibrosis Clinical Presentation"

_life, 2020, doi:10.3390/life11010014_

Round 1

Reviewer 1 Report

General comment

Bampi and colleagues reported the case of one patient with classical respiratory CF features and positive sweat tests values, carrying at the homozygous state the SNPs c.1584G>A (p.Glu528=) ; c.2562T>G (p.Thr854=) and c.4389G>A (p.Gln1463=); but no CF-causing variant following the whole CFTR locus sequencing. In an overexpressed mutated condition in heterologous in vitro cultured cells, authors showed that the c.(1584G>A;2562T>G;4489G>A) allele induced partial exon 11 skipping and intron 11 retention. Overexpression of the c.1584G>A sSNP into the CFTR cDNA in the CFBE41o- cells enhanced by up to 1.3-fold the protein expression level of the fully glycosylated functional CFTR form compared to wild-type CFTR. Total mRNA-sequencing (RNA-Seq) of patient-derived colorectal organoids did not show any aberrant splicing event. Of note, this innovative approach did not detect any change for genes implicated in the stress response categories, or tissue remodeling pathways as a signature of severe CF pathologies.

Some methodology aspects of the study are neglected and should be clarified. Important limitations of the study should also be mentioned. Conclusions on the putative pathogenic effect are overstated by authors in some sentences and required additional supporting evidences, or at least, a discussion about the hypothesis of an excess of CFTR protein and the links with the patient phenotype.

Comments in details :

Authors report two tiers of analyses for the patient: frequent severe disease-causing variants following a whole CFTR locus sequencing. The NGS strategy for the second tier, including metrics and variants have to be added in the manuscript. Authors should investigate c.1584G>A haplotype data and present variants in complex allele with this SNP in a supplemental material with splicing predictions of variants in regards to possible change in scores for canonical splicing sites but also regulatory splicing sequences (ISE).

The phenotype of the reported patient fulfills with the clinical definition of CF entity including respiratory features on scanner, positive sweat test and chronic pulmonary infections with opportunist microorganisms. Authors rightly mentioned that phenotype could rather correspond to atypical CF because of the patient presented no gastrointestinal symptoms, nutritional status was correct. However, considering the absence of any CF causing variant detected in this patient, its CF diagnosis , even if in a moderate form, remains highly debatable. Authors should discuss it, notably when regarding elements listed below. Onset of the disease for this was in childhood, but patient was only diagnosed with bronchiectasis at the age of 48 years. Early, recurrent and/or severe airway infection(s) (i.e. pertussis, or pneumonia) could contribute to the (moderate) decline of respiratory function of this patient. Whereas, no evident differential diagnosis was suggested by normal levels of immunoglobulins and nasal NO , the patient suffers for rheumatoid arthritis. Even if this chronic inflammatory disease was diagnosed 20 years later than the bronchiectasis diagnosis, the two pathologies could have common origin. Moreover, RNASeq of patient-derived colorectal organoids “did not detect any [change in ] genes from the stress response categories, i.e. inflammatory, unfolded protein response stress, or tissue remodeling pathways recently reported as a signature of severe CF pathologies”.

The case report could really become more informative if functional in vivo/ex vivo assays (Nasal Potential Difference or ionic current measurements on rectal biopsies) would be available for this patient. Functional tests are now crucial to demonstrate function effect of variant and/or confirm diagnosis. If no assay was performed in vivo/ex vivo, did authors perform swelling test on organoids? Is the basal luminal area of patient’s organoids different compared to organoids from CF subjects and healthy controls ? Did authors test CFTR modulators?

An abnormal swelling test or a specific response to modulators could be relevant to tip the scale in favor of CF diagnosis.

Authors reported in the abstract that c.1584G>A SNP is rare, however it is clearly wrong. The Variant allele Frequency in general population is 0.0167 (gnomAD V3 data) in average and this frequency could increase until 0.03 in some population. Even if not published, all specialized laboratories that performed extensive CFTR analysis worldwide (i.e. exons and flanking sequences analysis) have already detected this variant at the homozygous state among asymptomatic subjects (e.g. partners).

However, even its high frequency, the c.1584G>A is of particular interest, because of this location on the last nucleotide of exon 11. The “not neutral effect” of the variant was already suggested in previous studies especially in bronchiectasis because of the reduction of mRNA and protein level in heterologous in vitro systems (Bergougnoux et al., 2015, JCF).

Rather than CF-causing, this variant could appear as a susceptibility variant for epithelial diseases (pancreatic or respiratory) as suggested by previous reports (Casals, 2004, already referenced in the text [13]). A discussion about the hypothesis of an excess of CFTR protein as causative of a phenotype, possibly mild CF should also be added.

Some methodological aspects of experiments should be clarified.

mRNA splicing in transfected cells was evaluated without normalization with a housekeeping gene (Figure 1.A) that should be added.

The lack of reproductive data in RNASeq from organoids between R1 and R2 questions the relevance of compared data between patient and control. “Biological replicates” should be clarified in the manuscript because of two vials of cells for one patient only corresponds to technical replicates.

Author Response

Comments and Suggestions for Authors

General comment

Bampi and colleagues reported the case of one patient with classical respiratory CF features and positive sweat tests values, carrying at the homozygous state the SNPs c.1584G>A (p.Glu528=) ; c.2562T>G (p.Thr854=) and c.4389G>A (p.Gln1463=); but no CF-causing variant following the whole CFTR locus sequencing. In an overexpressed mutated condition in heterologous in vitro cultured cells, authors showed that the c.(1584G>A;2562T>G;4489G>A) allele induced partial exon 11 skipping and intron 11 retention. Overexpression of the c.1584G>A sSNP into the CFTR cDNA in the CFBE41o- cells enhanced by up to 1.3-fold the protein expression level of the fully glycosylated functional CFTR form compared to wild-type CFTR. Total mRNA-sequencing (RNA-Seq) of patient-derived colorectal organoids did not show any aberrant splicing event. Of note, this innovative approach did not detect any change for genes implicated in the stress response categories, or tissue remodeling pathways as a signature of severe CF pathologies.

Some methodology aspects of the study are neglected and should be clarified. Important limitations of the study should also be mentioned.  Conclusions on the putative pathogenic effect are overstated by authors in some sentences and required additional supporting evidences, or at least, a discussion about the hypothesis of an excess of CFTR protein and the links with the patient phenotype.

We are grateful to the Reviewer for their constructive comments and have accordingly edited the text to include more methodological details and revise the discussion on the pathological effect of the c.1584G>A sSNP.

Comments in details :

  1. Authors report two tiers of analyses for the patient: frequent severe disease-causing variants following a whole CFTR locus sequencing. The NGS strategy for the second tier, including metrics and variants have to be added in the manuscript. Authors should investigate c.1584G>A haplotype data and present variants in allele with this SNP in a supplemental material with splicing predictions of variants in regards to possible change in scores for canonical splicing sites but also regulatory splicing sequences (ISE).

Details on the NGS strategy of the patient sequencing is included in MM (see subsection 3.1, l. 223-235).

We are a little puzzled in understanding the comment of the Reviewer about investigating the haplotype in the complex allele. If the Reviewer suggests to predict the effect of c.1584G>A when also both other sSNPs are present – this is how we understand this comment – such analysis would not be possible. Each of the prediction tools assesses the effect locally, within the cognate exon/intron, while larger flanking sequences are not considered in the predictions. We performed the analysis for the other two sSNPs the patient is homozygous for. The analysis reveals no influence on the splicing pattern on their cognate exons, which also have been shown in the literature (see the Table below). We believe that including those predictions which do not report any influence on their cognate splicing junctions (and not at exon 11-intron 11 boundary) might be misleading and could be misinterpreted as if both sSNPs (c.2562T>G and c.4389G>A) ‘counteract’ the splicing effect on the c.1584G>A. Thus, we prefer to not add this information in Table 1, unless we misunderstood the comment of the Reviewer and they indeed mean something else.

Variant

Algorithm

NatGene2a

NNSplicea

Hbondb

MaxEntScan

ASSP

SpliceAI (Δ)a

c.1584G>A

0.89

0.95

12.6

6.42

9.53

0.04

wt

0.93

1

17.3

10.06

12.19

-

c. 2562T>G

-

-

8.0

-4.57

-

0.0

wt

-

-

1.8

-19.82

-

-

c. 4389G>A

-

-

-

-18.53

-

0.0

wt

-

-

-

-9.16

-

-

  1. The phenotype of the reported patient fulfills with the clinical definition of CF entity including respiratory features on scanner, positive sweat test and chronic pulmonary infections with opportunist microorganisms. Authors rightly mentioned that phenotype could rather correspond to atypical CF because of the patient presented no gastrointestinal symptoms, nutritional status was correct. However, considering the absence of any CF causing variant detected in this patient, its CF diagnosis, even if in a moderate form, remains highly debatable. Authors should discuss it, notably when regarding elements listed below. Onset of the disease for this was in childhood, but patient was only diagnosed with bronchiectasis at the age of 48 years. Early, recurrent and/or severe airway infection(s) (i.e. pertussis, or pneumonia) could contribute to the (moderate) decline of respiratory function of this patient. Whereas, no evident differential diagnosis was suggested by normal levels of immunoglobulins and nasal NO , the patient suffers for rheumatoid arthritis. Even if this chronic inflammatory disease was diagnosed 20 years later than the bronchiectasis diagnosis, the two pathologies could have common origin. Moreover, RNASeq of patient-derived colorectal organoids “did not detect any [change in ] genes from the stress response categories, i.e. inflammatory, unfolded protein response stress, or tissue remodeling pathways recently reported as a signature of severe CF pathologies”.

As correctly acknowledged by the Reviewer the patient presents CF symptoms (e.g. respiratory features on scanner, positive sweat test and chronic pulmonary infections), but lack gastrointestinal symptoms, thus we uniformly use not throughout the text ‘atypical CF phenotype’. The motivation for the whole study was exactly this controversy between some classical CF symptoms the patient has on one hand and the lack of known CF-causing mutation along with the advanced age of diagnosis on the other. The NGS sequencing of the whole CFTR locus revealed variants in a specific unique and rare constellation in this patient, neither of which alone is considered a CF-causing. However, our molecular analysis provides an evidence that the epistatic interplay between these three sSNPs is likely to explain the very mild atypical disease phenotype. Reflecting on the comments of the Reviewer we edited at many places the text so that we (1) leave no impression that we would claim that even homozygous c.1584G>A sSNP is disease-causing, and (2) emphasize on the atypical CF phenotype. We also extended the discussion with published reports for the effect on the c.1584G>A sSNP on the mRNA and protein level in in vitro cell culture, arguing against the neutrality of this mutation on CFTR function.

The Reviewer mentions that the RNA-Seq does not capture any gene expression alterations of genes from the inflammatory, unfolded protein response stress, or tissue remodeling pathways which we recently reported as a signature of severe CF pathologies (J Cyst Fibr. 2020). We wish to emphasize that we use organoids as an infection-free environment and thus allow for detecting solely mutation-driven expression changes. In contrast, primary CF HBE cells from lung biopsies comprise the complex signature of mutation-driven CFTR unfolding along with chronic bacterial infections and massive inflammatory responses commonly contributing to CF pathogenesis (Bampi et al, J Cyst Fib. 2020, ref 18 in the manuscript). We edited the text to clearly state this specific feature, the infection-free origin of the organoids, so that by sequencing patient-derived organoids we can disentangle the mutation-driven effect only. The changes are l. 166-168: “Unlike lung-derived material, colorectal organoids are derived from an infection-free environment and are thus suitable to address whether a mutation-triggered changes in CFTR expression alone elicit stress response.”

  1. The case report could really become more informative if functional in vivo/ex vivo assays (Nasal Potential Difference or ionic current measurements on rectal biopsies) would be available for this patient. Functional tests are now crucial to demonstrate function effect of variant and/or confirm diagnosis. If no assay was performed in vivo/ex vivo, did authors perform swelling test on organoids? Is the basal luminal area of patient’s organoids different compared to organoids from CF subjects and healthy controls? Did authors test CFTR modulators? An abnormal swelling test or a specific response to modulators could be relevant to tip the scale in favor of CF diagnosis.

We did not perform swelling assay, since the organoids are already swollen at baseline without adding of forskolin. The basal luminal area of the patient’s organoids is similar to the non-CF controls as we referred to in the paper. There was a visible lumen filled with liquid reflecting the minimal necessary CFTR function to show a normal swelling of the organoids at baseline. We did not test CFTR modulators. We added some details in l. 87-88: “and at baseline were already swollen with basal luminal area similar to the non-CF controls.”

  1. Authors reported in the abstract that c.1584G>A SNP is rare, however it is clearly wrong. The Variant allele Frequency in general population is 0.0167 (gnomAD V3 data) in average and this frequency could increase until 0.03 in some population. Even if not published, all specialized laboratories that performed extensive CFTR analysis worldwide (i.e. exons and flanking sequences analysis) have already detected this variant at the homozygous state among asymptomatic subjects (e.g. partners).

However, even its high frequency, the c.1584G>A is of particular interest, because of this location on the last nucleotide of exon 11. The “not neutral effect” of the variant was already suggested in previous studies especially in bronchiectasis because of the reduction of mRNA and protein level in heterologous in vitro systems (Bergougnoux et al., 2015, JCF).

We thank the Reviewer for this comment. We believe that we left a wrong impression of claiming the mutation is rare which might be from somewhat unprecise writing. Indeed, the c.1584G>A sSNP is relatively frequent as heterozygous sSNP, but rare in homozygosity. We have edited the text (abstract, introduction) to clearly state this and cite thereby the gnomAD V3.1 data base.

The suggested citation (Bergougnoux et al., 2015, JCF) is also included and the results of this paper mentioned in the introduction and in the discussion.

  1. Rather than CF-causing, this variant could appear as a susceptibility variant for epithelial diseases (pancreatic or respiratory) as suggested by previous reports (Casals, 2004, already referenced in the text [13]). A discussion about the hypothesis of an excess of CFTR protein as causative of a phenotype, possibly mild CF should also be added.

We have edited the discussion and throughout the text to consistently use atypical and very mild CF phenotype. See also our responses to comment 2 and the edited discussion.

  1. Some methodological aspects of experiments should be clarified. mRNA splicing in transfected cells was evaluated without normalization with a housekeeping gene (Figure 1.A) that should be added.

We present a qualitative and not a quantitative analysis of the effect of c.1584G>A SNP on the CFTR mRNA (Fig. 1A). We agree with the Reviewer that for a quantitative analysis a normalization with housekeeping gene. Please note that we aim on qualitative analysis and thus did quantify the amplicons. The non-quantitative aspect of the approach we use is mentioned in the Methods section (l. 244-245) and Fig. 1 caption (l. 116)

  1. The lack of reproductive data in RNASeq from organoids between R1 and R2 questions the relevance of compared data between patient and control. “Biological replicates” should be clarified in the manuscript because of two vials of cells for one patient only corresponds to technical replicates.

As established in the RNA-Seq-based approaches, technical replicates would be libraries prepared in duplicates from the same organoid. Our replicates were prepared from independently cultured organoids and are thus, clearly not technical. Indeed, they fall into the broad category of biological replicate however from the same donor. Reflecting on the Reviewer’s comment we re-named the replicates to ‘two independent replicates’ (l. 145, 146 and l. 254).

We also kindly disagree with the Reviewer that the reproducibility between both replicates R#1 and R#2 is low. We represent the Pearson coefficient as squared value which actually is the more accurate way (R2=0.92) than the more often used R which would then be 0.96.

Reviewer 2 Report

In this manuscript entitled "The effect of synonymous single-nucleotide polymorphism on an atypical cystic fibrosis clinical presentation" the authors report a patient who was diagnosed with CF at an advanced age and present very mild CF symptoms. A rare homozygous sSNP (c.1584G>A) at the end of exon 11 in the CFTR gene domain was highlighted causing cognate exon skipping and retention of a sequence from the downstream intron, both of which however occur at a relatively low frequency. Two other sSNPs (c.2562T>G (p.Thr854=) and c.4389G>A (p.Gln1463=)), for which the patient is also homozygous were also found as to stabilize the CFTR protein expression and compensating for partially inefficient splicing.

This is a well written manuscript of a high standard. The paper should be accepted in its present form. No modification is necessary.

Author Response

We were very pleased to read the assessment of the Reviewer.

Reviewer 3 Report

The authors present a very interesting case report arguing for pathogenic impact of three synonymous variants in the CFTR gene. 

Overall the manuscript is well considered, well written and methodologically sound. It mainly opens novel reaserch avenues.

I have only one minor comment, in that the authors could present also results from the SpliceAI bioinformatic algorithm, which is increasingly used in this context. 

Author Response

We greatly thank the Reviewer for their supporting comments. Reflecting on the Reviewer’s comment, we included in Table 1 the predictions of SpliceAI algorithm. The results corroborate the predictions of all other algorithms.

Reviewer 4 Report

This is very important manuscript that shows that SNPs are functional. Authors perform solid and very rare in this field  work that shows the mechanistic consequences of CFTR SNP. Considering that the workflow included NGS, organoids cultures and biochemical and clinical assessment, I believe these studies are solid and of high value.

As for minor comments:

  1. Would author try to compare their patient donor data with analogical records (if available in CF database)?
  2. DeltaF508 leads also to synonymous mutation in CFTR ILE – please see doi: 1074/jbc.M110.154575
  3. SNPs can also affect miRNA binding

Author Response

We greatly appreciate the Reviewer’s positive feedback.

  1. The Reviewer raises a very important question which unfortunately with the current data that are available is not possible to address. Since the c.1584G>A sSNP is generally considered as non-CF causing, this mutation is not included in the CF data base (cftr2), hence no comparison with other CF patients is possible. Usually, the CFTR gene of individuals with clear CF clinical presentation is only punctually sequenced for a set of common CF mutations (all well-documented in cftr2 data base). Unfortunately, the full genetics of the patients is largely missing. The growing experimental evidence on the epistatic effects of non-disease-causing variations and likely their contribution to disease heterogeneity and theratyping launched the whole-exome sequencing project of CF patents funded by the Cystic Fibrosis Foundation; the data of which would be the perfect set to address Reviewer’s question are not publicly available yet. This data set would definitely enable new insights into the complex interactions of mutations (both within the CFTR gene and with other genes) and consequently in shaping the complex and heterogenous CF clinical phenotype.
  2. and 3. We have included in the introduction the effect of sSNPs on miRNA-mediated gene regulation including two representative publications.

The cited publication (doi: 10.1074/jbc.M110.154575) is already in our reference list and the results from this publication are discussed when summarizing the effect of sSNPs, i.e. the effect of this sSNP is on the secondary structure.

Round 2

Reviewer 1 Report

Authors well answered to all comments. The manuscript is now suitable for publication in the journal.